# Education and Information to Improve Adherence to Screening for Breast, Colorectal, and Cervical Cancer—Lessons Learned during the COVID-19 Pandemic

**DOI:** 10.3390/cancers16173042

**Published:** 2024-08-31

**Authors:** Raimondo Gabriele, Monica Campagnol, Paolo Sapienza, Valeria Borrelli, Luca Di Marzo, Antonio V. Sterpetti

**Affiliations:** Department of Surgery, Policlinico Umberto I, Sapienza University, Viale del Policlinico, 00167 Rome, Italy; raimondo.gabriele@uniroma1.it (R.G.); paolo.sapienza@uniroma1.it (P.S.); v.borrelli@policlinicoumberto1.it (V.B.); luca.dimarzo@uniroma1.it (L.D.M.)

**Keywords:** trends mortality rates, cancer, COVID-19 pandemic, screening

## Abstract

**Simple Summary:**

Screening for breast, colorectal, and cervical cancer is correlated with diagnosis at an earlier stage, less extensive surgery, and reduced mortality and fewer complications. Adherence rates to cancer screening are lower for individuals with low socio-economic conditions and educational attainment. These social disparities are only partially reduced by free screening through national initiatives. Education and information and appropriate expenditure for preventive care have the potentials to increase adherence to screening for colorectal, breast, and cervical cancer with the possibility of reduced cancer mortality. The findings of our study highlight the importance of the implementation of nationally organized screening programs for several other types of cancers that are often detected after the occurrence of symptoms. Nationally organized screening programs for several types of cancers, like esophageal, gastric, and pancreatic cancer, in regions with a high prevalence may increase the possibility of diagnosis at earlier stages and improved early and late results.

**Abstract:**

The objective of this study was to determine the correlation between adherence to cancer screening programs and earlier diagnosis of the 14 most common types of cancers in the adult population, before and during the COVID-19 pandemic. National data concerning number of admissions and operations in Italy for adult patients admitted with oncologic problems during the COVID-19 pandemic (2020 to 2022) and in the pre-pandemic period (2015 to 2019) were analyzed. We selected 14 types of cancer that present the most common indications for surgery in Italy. This study included 1,365,000 adult patients who had surgery for the 14 most common types of cancer in the period 2015–2022, and interviews concerning adherence rates to screening for breast, colorectal, and cervical cancer were conducted for 133,455 individuals. A higher decrease in the number of operations for the 14 types of cancer (−45%) was registered during the first three acute phases of the pandemic, and it was more evident for screenable cancers like breast, colorectal, and cervical cancer (*p* < 0.001). During the first year of the COVID-19 pandemic, the number of screened individuals for breast, colorectal, and cervical cancer decreased by 33.8% (from 7,507,893 to 4,969,000) and the number of diagnoses and operations for these three types of cancer decreased by 10.5% (from 107,656 to 96,405). The increase and return to normality of the number of screened individuals in the last year of the pandemic (2022) and in the first post-pandemic year (2023) was associated with a return to the pre-pandemic levels of diagnoses and operations. The adherence rates were lower for individuals living in rural areas, with low socio-economic status, and unmarried persons; however, the most statistically significant factor for reduced adherence was a lower level of educational attainment. Free screening through nationally organized programs reduced social disparities. There were no significant differences between the pre-pandemic and pandemic periods for several types of cancers (stomach, esophagus, pancreas, liver) that are diagnosed for the occurrence of symptoms and for which nationally organized programs might increase the possibility of earlier diagnosis and improved clinical outcomes. Education, information, and appropriate expenditure for preventive care have the potential to reduce cancer mortality. Nationally organized screening programs for several types of cancers, which are often detected for the occurrence of symptoms, may increase the possibility of diagnosis at earlier stages.

## 1. Introduction

The COVID-19 pandemic has brought reduced hospital/clinic visits and surgical procedures. During the first year of the pandemic, the number of diagnoses of cancers decreased by 10.5%; in the lockdown periods, the decrease reached 50% [1,2,3,4,5]. Screening of the general population has been correlated with decreased mortality for breast, colorectal, and cervical cancers. Organized screening has been associated with reduced mortality in Europe. Cardoso et al. [6] and Ola et al. [7] found that organized screening programs for colorectal cancer with fecal tests, especially when all eligible groups were covered, achieved the highest utilization of the screening tests, with favorable survival rates for patients with screen-detected colorectal cancer. These favorable results were also seen within each stage—the five-year overall survival rates for patients with screen-detected stage I, II, III, and IV cancers were 92.4% (95% CI 91.6–93.1), 87.9% (86.6–89.1), 80.7% (79.3–82.0), and 32.3 (29.4–35.2), respectively. The possibility that screening by sigmoidoscopy may reduce colorectal cancer prevalence and mortality has been supported by Brenner et al. [8,9]. The association between routine screening and reduced mortality for breast and cervical cancer has become standard evidence [10,11]. It may appear unethical, but several lessons can be derived from the COVID-19 pandemic, including the importance of screenings for breast, colorectal, and cervical cancer and the negative consequences of their decrease.

The aim of our study was to analyze the number of admissions to hospital, the number and type of surgical procedures, and the results for patients with oncologic problems, during the pandemic years (2020 to 2022) in Italy. We analyzed the number of surgical procedures performed for 14 common types of cancer in Italy. These data were compared to those of the pre-pandemic period (2015–2019). This study aimed to confirm the importance of cancer screening and to identify subgroups of patients with lower adherence rates to screenings.

## 2. Methods

### 2.1. Study Design

We analyzed the data concerning the number of admissions and operations in Italy for patients admitted with oncologic problems during the COVID-19 pandemic (2020 to 2022) and in the pre-pandemic period (2015 to 2019). We selected 14 types of cancer that present the most common indications for surgery in Italy.We analyzed the mortality rates in Italy during the pre-pandemic period (2015–2019) and in the pandemic period (2020–2022) related or not to COVID-19 infection.A comparison was made between the characteristics (number/adherence) of preventive screening for breast, colorectal, and cervical cancer during the pre-pandemic and the pandemic periods.

The data about point 1 were extracted from the PNE (Programma Nazionale Esiti) reports of the AGENAS (Agenzia Nazionale per i Servizi Sanitari Regionali) [12]. AGENAS is a non-profit agency, part of the National Health Minister, supported by funds from the Italian Government, which includes data from 1377 public and private hospitals, including 97% of hospital admissions in Italy.

The data about point 2 were extracted from the reports of the ISTAT (Italian Institute for Statistics), ISS (Istituto Superiore Sanità), and EUROSTAT [13,14,15].

ISTAT, ISS, and EUROSTAT are non-profit governmental public institutions. The reports describe the mortality rates in Italy and in the European Union during the pre-pandemic and pandemic periods. 

The data about point 3 were based on the report of Italian National Institutions concerning cancer and screening [15,16,17].

All the patients gave written approved consent.

### 2.2. Data Analysis

The primary outcomes were changes in the number and type of operations performed. The secondary outcomes were the 30-day mortality and morbidity rates after surgery.

IRB approval was not required for this study. All the patients gave written approved consent.

### 2.3. Statistical Analysis 

The categorical variables are expressed as frequencies and percentages. The continuous variables with normal distribution are expressed as mean and standard deviation. The Student’s *t* test and the chi square test were used where appropriate.

## 3. Results

### 3.1. Excess Mortality during the Pandemic Period (2020–2022)

During the three years of the pandemic, an excess mortality of 251,911 was registered. In Italy, 73% of the excess deaths were related with the COVID-19 infection and 27% occurred in patients who were COVID-19 negative. A similar trend was reported for all countries in the European Union. The data about the more common reported causes for excess of mortality for patients who were COVID-19 negative are available only for the year 2020. In Italy, during the year 2020, the age-standardized mortality rate (×100,000) for cardiovascular deaths was 281.1; for cancers, it was 227.0, and it was 100.7 for COVID-19. In the year 2023, there was no excess mortality in Italy or in the European Union.

### 3.2. Reduced Number of Hospital Admissions and Surgeries for the 14 Analyzed Types of Cancer

Table 1 shows the number of operations performed for the 14 analyzed types of cancer in Italy. In the year 2020, 160,617 operations for the 14 analyzed types of cancer were performed in Italy with a decrease of 10% (−18,000 operations) in comparison with the expected number considering the trends from the previous five years. In the year 2021, a decrease of 4% was registered (−6000 operations), whereas, in the last year of the pandemic (2022), the number of operations was almost similar to that in the pre-pandemic period.

A higher decrease in the number of operations for the 14 types of cancer (−45%) was registered during the first three acute phases of the pandemic (March–April 2020; October–December 2020; January–March 2021) in which there was almost a complete cancellation of screening and follow-up oncologic visits and reduced hospital admissions.

### 3.3. Heterogeneous Decrease in Surgical Operations

#### 3.3.1. First Year of the Pandemic (2020)

The significant decrease of −10% for oncologic surgical interventions during the first year of the pandemic (2020) had a statistically significant heterogeneous distribution (*p* < 0.001).

The number of operations was similar in the pre-pandemic and post-pandemic periods for specific pathologies, which are often diagnosed because of the occurrence of severe symptoms and for which systematic screening is not performed in Italy. Surgery for patients with cancer of the esophagus, pancreas, bladder, and ovary is often diagnosed by investigative techniques to identify the cause of evident symptoms (dysphagia, jaundice, hematuria, abdominal pain). This evidence implies that the medical system answered promptly to the attention and worries of the patients, despite the inevitable difficulties related with the prevention of the diffusion of the pandemic. 

There was a statistically significant decrease in admissions and operations for pathologies that are more often discovered in asymptomatic patients by screening programs, like breast cancer, colon and rectal cancer, prostate cancer, and cervical cancer (Table 1).

Table 2, Table 3, Table 4, Table 5, Table 6 and Table 7 show the reduced number of asymptomatic patients who attended organized screening programs in Italy. 

We also documented a statistically significant reduction (*p* < 0.05), even if less evident, in operations related with pathologies that are often diagnosed because of less vague symptoms, implying a diagnostic course, which might have been deferred because of the generalized fear of contamination and the tendency to defer diagnostics for less invalidating symptom. Screening programs are not systematically performed in Italy for these pathologies. 

#### 3.3.2. Second and Third Year of the Pandemic (2021–2022)

During the second and third year of the pandemic, hospital admissions and surgical procedures for patients with cancers that are also detected by well-organized and planned screening programs (breast, colon–rectum, cervical cancer) returned slowly to the pre-pandemic levels, simultaneously with the return to almost-normal medical practice.

The number of admissions and surgical procedures for pathologies associated in general with milder symptoms, which address a diagnostic course with radiological and endoscopic tests, had a statistically significant decrease to reach the pre-pandemic levels (*p* < 0.05).

Operations for patients with gallbladder cancer decreased significantly during the three years of the pandemic, simultaneously with the statistically significant reduction in open and laparoscopic cholecystectomies. Most diagnoses of gallbladder cancer are made after a pathological examination of the gallbladder that was removed due to the presence of stones. 

#### 3.3.3. Adherence to Screening for Breast, Colorectal, and Cervical Cancer

Attendance to screening decreased by 33.8% during the first year of the pandemic with reduced numbers of diagnoses of cancers and adenomas: cancers were diagnosed at a more advanced stage during the first year of the pandemic (Appendix A). Adherence to screening returned to the pre-pandemic levels during the second and third years of the pandemic (2021–2022). Several factors were found to be correlated with reduced adherence to screening, including unmarried status, younger age, living in rural areas, and low economic status. Multiple regression analysis demonstrated that the most important factor influencing attendance at cancer screening was the level of educational attainment and the frequency of consultation with family doctors (Table 3, Table 5 and Table 7). These disparities were partially reduced by free screening organized by the Italian National System. Despite significant differences related to the socio-economic status, the attendance rates were inappropriately low either in the pre-pandemic or in the pandemic period.

#### 3.3.4. Comparison with Other European Countries

Similarly to the case in Italy, in other EU countries, several factors were found to be correlated with reduced adherence to screening, including unmarried status, younger age, living in rural areas, low economic status. Again, the most significant factors were level of educational attainment, expenditure for preventive care, and education and counseling, including frequency of consultation and availability of family doctors (Figure 1, Figure 2 and Figure 3).

Increased adherence to screening is significantly related to the implementation of organized programs. Considering only screening for colorectal cancer, the rate of participation in screening programs differs significantly across European countries. The lowest participation is observed in countries in which organized, population-based screening programs are not yet implemented. In 2019, 49% of people aged 50 to 74 reported that they had never attended screening for colorectal cancer, with a lack of participation as high as 94% in some countries and as low as 17% in others (Figure 2). The incidence of colorectal cancer has decreased significantly in the last years (2000–2017) in countries with long-standing screening programs and widespread population coverage, while it either remained stable or increased in countries with no large-scale screening programs (Appendix A). 

In all European countries, the most significant correlation between adherence to screening was related with the level of expenditure for preventive care, education, and counseling (Figure 4). Expenditure for preventive care, education, and counseling imply several positive actions, including the prevention of obesity, control of diabetes, appropriate diet, and implementation of adequate cancer screening programs. The expenditure for preventive care allows national and local initiatives, addressing the importance of screening, and an adequate number of counseling centers and a higher number and availability of family doctors.

## 4. Discussion

During the pandemic period, namely during the acute phases, deferrable surgeries were rarely performed, and patients asked for medical advice only for severe symptoms and were subsequently operated on, if needed. As consequence, there was an increased number of emergency operations, for cancers diagnosed at a more advanced stage [1,2,3,4,5].

The pandemic confirmed the importance of organized screening programs for cancer of the breast, colon–rectum, and cervix [18,19,20,21,22]. Reduced screening was associated with a lower number of diagnoses and operations, as well as with the diagnosis of cancer at more advanced stages. Reduced adherence to screening programs included all levels of the population, and it was more evident for people with lower educational attainment, those with a lower family income, and immigrants from low-income countries [23,24,25]. Timely treatment may have been less accessible to vulnerable patient populations.

Free screening through the Italian National System reduced social disparities either before or during the pandemic. However, still, the overall adherence rates were lower than expected even in individuals with higher educational attainment and family income, and social disparities persisted despite the free screening [26,27].

Proper education and information about the importance of preventive care may increase the adherence to cancer screening programs. Valid information requires several forms of communication including general and location-specific considerations. The first form of communication should be a commitment from national institutions, including mass-media campaigns and teaching in schools, universities, and workplaces; the second aspect, probably the most difficult and effective, should be reserved to local communities, including clinicians, nurses, and small hospitals. Family doctors play a major role in this setting. Almost 90% of the individuals with low educational attainment and a lower family income attended screening because of the advice of family doctors.

In this context, a close collaboration between policy makers, health care providers and physicians is fundamental, assuring a good cost-effectiveness ratio for health spending.

Another observation relates to the similar number of operations performed before and during the pandemic for specific forms of cancer for which an organized screening program in Italy has not yet been implemented. This evidence supports the concept that, for these types of cancers, the diagnosis is based mainly on the occurrence of symptoms, with a consequent delay in treatment and diagnosis at later stages [28,29,30,31]. Thus, it is conceivable to introduce screening in Italy for specific types of cancers in regions with high prevalence of the disease. Screening for gastric cancer and esophageal cancer has led to earlier diagnosis, less invasive treatment, and improved survival rates in several East Asian countries.

## 5. Conclusions

Education and information and appropriate expenditure for preventive care have the potential to increase adherence to screening for colorectal, breast, and cervical cancer with the possibility of reduced cancer mortality. The findings of our study highlight the importance of the implementation of nationally organized screening programs for several other types of cancers that are often detected after the occurrence of symptoms. Nationally organized screening programs for several types of cancers, like esophageal, gastric, and pancreatic cancer, in regions with a high prevalence may increase the possibility of diagnosis at earlier stages an improved early and late results.

## Figures and Tables

**Figure 1 cancers-16-03042-f001:**
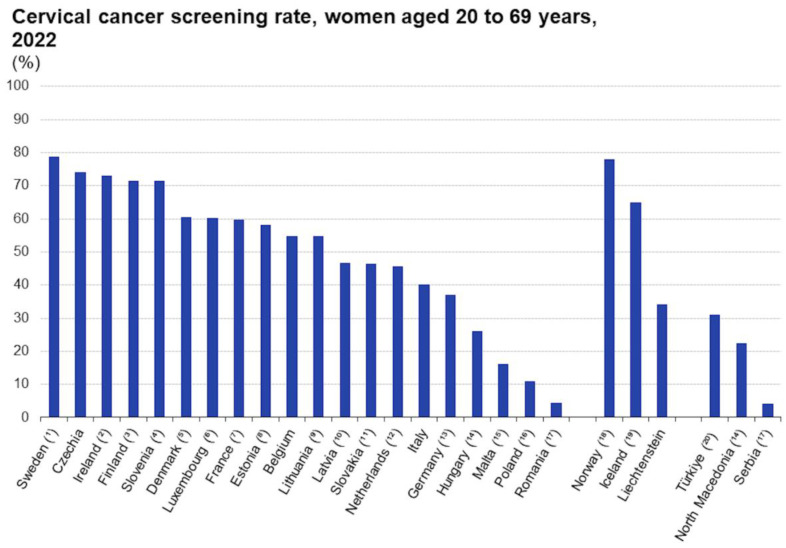
Adherence rates to cervical cancer screening in Europe.

**Figure 2 cancers-16-03042-f002:**
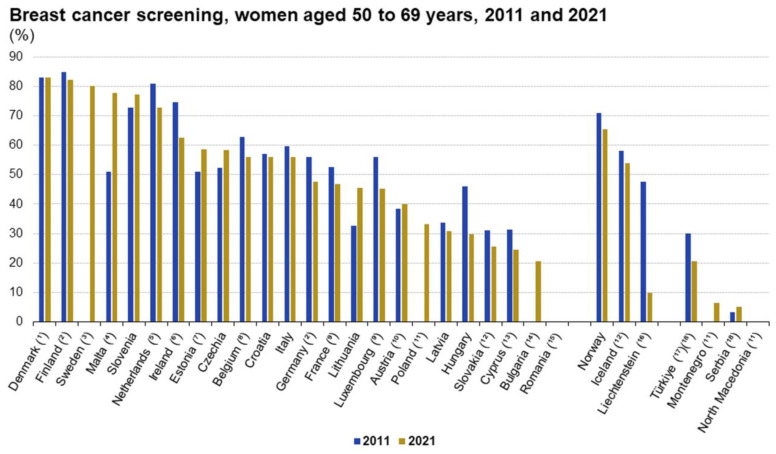
Adherence rates to mammogram breast cancer screening in Europe.

**Figure 3 cancers-16-03042-f003:**
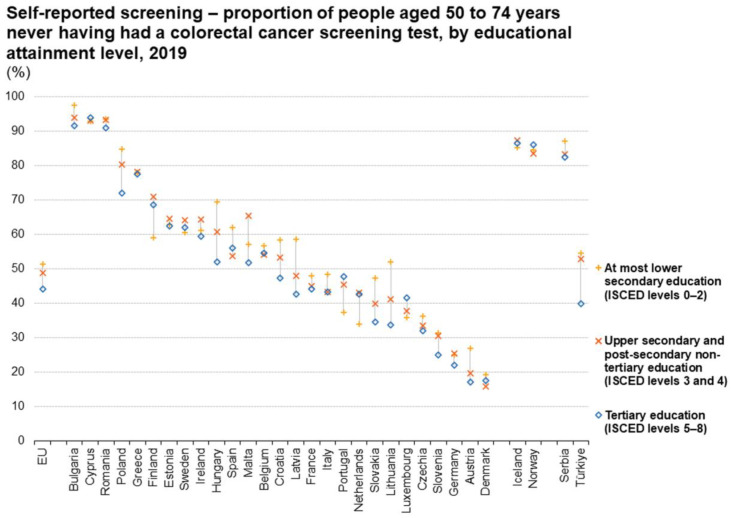
Adherence rates in Europe to colorectal cancer screening.

**Figure 4 cancers-16-03042-f004:**
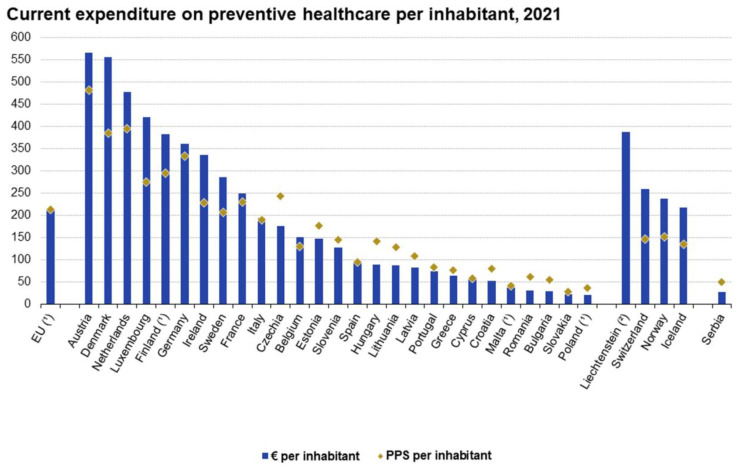
Expenditure for preventive care, education, and counseling in Europe (2012–2020). The higher the level of expenditure for preventive care, education, and counseling, the higher the adherence rates to cancer screening.

**Table 1 cancers-16-03042-t001:** Number of patients who had surgery for 14 types of cancers in the pre-pandemic period (2015–2019) and in the pandemic period (2020–2021–2022) in Italy.

	2015	2016	2017	2018	2019	Pre-Pandemic Mean Annual	2020	2021	2022	Pandemic Mean Annual (% Difference) *
Esophagus	795	843	789	831	856	823	827	869	883	860 (+4.5%)
Stomach	6746	6557	6239	6146	5824	6302	5088	5075	4890	5018 (−19%)
Gallbladder	890	917	840	837	771	851	713	731	736	727 (−9%)
Liver	6408	6392	6303	6352	6610	6413	6195	5961	6126	6094 (−7.7%)
Pancreas	2626	2648	2690	2809	2710	2697	2778	2766	2938	2827 (+2%)
Colon	27,019	26,784	26,849	27,127	26,233	26,802	23,078	24,796	25,542	24,472 (−9.4%)
Rectum	7212	6844	6679	6668	6051	6691	5627	5615	5685	5642 (−6.7%)
Kidney	10,935	11,002	11,129	11,736	11,907	11,342	10,665	11,676	12,481	11,607 (−3.1%)
Prostate	18,952	18,972	18,673	20,270	20,688	19,511	17,115	18,645	21,324	19,028 (−9.4%)
Bladder	5294	5302	5337	5201	5211	5269	5241	5101	5037	5126 (−1.4%)
Breast	60,630	62,172	61,797	62,738	62,343	61,936	56,057	62,764	63,986	60,935 (−2.5%)
Lung	11,454	11,590	12,039	12,458	12,782	12,065	11,637	12,083	12,808	12,176 (−6.4%)
Uterus	11,743	12,036	12,044	11,961	12,349	12,027	11,643	12,103	12,106	11,951 (−4.4%)
Ovary	4004	4042	3937	3978	4058	4004	3953	3909	4100	3987 (−2.4%)
TOTAL	174,708	176,101	175,345	179,112	178,393		160,617 (−10%) *	172,094 (−4%) *	178,642 (−0.5%) *	

* The % mean difference with the expected number of operations on the basis of the trend in the five years before the pandemic.

**Table 2 cancers-16-03042-t002:** Results of the national organized screening program for breast cancer in Italy before (2018–2019) and during the pandemic (2020–2021).

	2018	2019	2020	2021
Population of Italian women 50–69 years	8,695,338		8,670,039	
Women 50–59 years	4,842,322		4,803,491	
Women 60–69 years	3,853,016		3,866,548	
Invitations	3,364,979	3,582,635	2,593,288	3,569,765
Attendance (%)	60.5	60.7	51.0	56.2
Number of mammograms	1,822,8511	1,876,721	1,242,415	1,937,375
Benign lesions	1036	952	741	1343
In situ ductal carcinoma	1069	1135	801	1316
Invasive carcinoma	8045	8300	6061	9845
Invasive carcinoma < 1 cm diameter	2409	2455	1781	2786

**Table 3 cancers-16-03042-t003:** The % of interviewed women who had a mammogram screening within the last two years inside and outside the national organized system. Interviews made before (2017–2019) and during the pandemic period (2020–2021) in Italy.

Total Interviewed Women 28.072	Attendance to Screening Programs	
	2017–2019	2020–2021
	**Total** **75%**	**Total** **71%**
AGE 50–59	77%	73%
AGE 60–69	73%	68%
Educational level *		
1	65%	50%
2	73%	67%
3–4	78%	75%
5–8	82%	80%
Low income	60%	55%
Middle income	72%	65%
Good income	81%	77%
Italians	75%	72%
Non-Italians **	70%	59%

* Educational level is based upon the International Standard Classification of Education (ISCED), 1997 version, and refers to 1—pre-primary, primary education; 2—lower secondary education; 3–4—upper secondary and post-secondary non-tertiary education; 5–8—tertiary education. ** Immigrants from low-income countries.

**Table 4 cancers-16-03042-t004:** Results of national screening program screening for colorectal cancer in Italy before (2018–2019) and during the pandemic (2020–2021).

	Colorectal Cancer Screening—Fecal Occult Blood Test			
	2018	2019	2020	2021
Italian population 50–69 years of age	17,123,098		17,233,176	
Number invited	5,939,182	5,921,032	4,159,765	6,416,162
Attendance (%)	42.7%	40.5%	34.1%	38.7%
Number of tests	2,570,437	2,619,871	1,487,636	2,607,329
% of positive tests	4.8%	5.0%	5.5%	5.0%
No. of colonoscopies after positive test	97,604	105,592	60,754	99,100
Diagnoses of carcinoma	2418	2877	1402	2678
Diagnoses of large adenoma	14,870	17,356	10,286	16,000
% of endoscopic resections	17.9%	14.6%	19.3%	18.2%

**Table 5 cancers-16-03042-t005:** Screening for colorectal cancer (interviews of 51.706 persons aged 50–69 years).

Subgroups with Statistical Significance at Univariate Analysis	% of Interviewed Persons Who Had Timely Screening	Inside the National Organized Screening Program	Outside the National Organized Screening Program
	47%	39%	8%
Sex			
Males	48%	39%	8%
Females	46%	39%	7%
Age			
50–59	42%	34%	7%
60–69	52%	44%	8%
Education			
1	37%	31%	6%
2	45%	39%	6%
3–4	49%	41%	8%
5–8	50%	38%	12%
Family income			
Low	32%	26%	6%
Middle	41%	34%	7%
High	54%	46%	8%
Citizenship			
Italian	47%	39%	8%
Immigrants (low-income countries)	43%	39%	4%

**Table 6 cancers-16-03042-t006:** Results of National Organization Screening Program for cervical cancer in Italy before (2018–2019) and during the pandemic (2020–2021).

Screening Test	2018	2019	2020	2021
No. of Italian Women Aged 25–64 Years	16,183,088	16,190,022	16,191,04	16,232,654
Overall invited women for screening	3,966,409	3,835,318	2,598,295	3,426,660
Adherence to invitation (%)	41.7	40.7	34.2	39.2
Cytology	2,453,583	2,212,192	1,223,873	1,434,395
Adherence to invitation (%)	34.2	33.7	27.5	34.8
Diagnoses of CIN2+ for every 1000 examined women	4.6	5.0	5.5	4.5
HPV Test + Cytology	1,480,776	1,632,362	1,360,553	NA
Adherence to invitation (%)	48.5	45.2	37.6	NA
Diagnoses of CIN2+ for every 1000 examined women	5.6	6.4	6.8	NA
Number of total diagnoses for CIN2+	7.177	7.625	5.228	NA

**Table 7 cancers-16-03042-t007:** Screening for cervical cancer (cytology or HPV + CYTOLOGY) (interviews of 53.677 women aged 25–64 years).

Subgroups with Statistical Significance at Univariate Analysis	% of Women Who Had Timely Screening	Inside the National Organized Screening Program	Outside the National Organized Screening Program
	79%	49%	30%
Age			
25–34	74%	42%	32%
35–49	83%	47%	36%
50–64	78%	53%	25%
Education			
1	60%	43%	17%
2	74%	51%	23%
3–4	81%	52%	31%
5–8	84%	47%	37%
Family Income			
Low	67%	44%	23%
Middle	77%	48%	29%
High	84%	52%	32%
Citizenship			
Italian	79%	48%	31%
Immigrants (low-income countries)	74%	55%	19%

## Data Availability

Data are available on request by antonio.sterpetti@uniroma1.it.

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
