# Peer review of "Education and Information to Improve Adherence to Screening for Breast, Colorectal, and Cervical Cancer—Lessons Learned during the COVID-19 Pandemic"

_cancers, 2024, doi:10.3390/cancers16173042_

Round 1

Reviewer 1 Report

Comments and Suggestions for Authors

Title: Education and information to Improve Adherence...in reality a comparison is being made Before and After the suspensions of activities linked to the COVID-19 pandemic.

Abstract: the only data reported is the 45% decrease in cancer operations: nothing for tumor site and all other information reported in the table.

Introduction: where is the introduction? The authors report 6 references: is it therefore intended that the reader downloads the 6 articles, reads them and makes his own personal introduction?

The aim of the study: report the aims indicated here with the same sequence as Study design and in the Results where the first item described is mortality.

Line 73: data were based on ...references 11, 12, 13, 14 ...we need to read what they report.

Line 74: All patients gave written approved consent: how did you obtain informed consent using these sources?

Line 80: IRB: clarify acronym; All patients gave written approved allow: again?

Line 88: mortality data: but the first objective is to analyze the number of admissions to hospital.

Line 90: Counties? Maybe countries. This comment is not clear between Italy and European countries and not between Italy and previous years. What does data mean for CVD, cancer and Covid? Compared to what?

Line 96-97-98: repeated 3 times you have analyzed 14 tumor sites.

Table 1: only the last line is commented on (-10%, -4% and -0.5%): what can we say about the single tumor sites analysed? Specify that here by uterus we mean the corpus of the uterus and not the uterine cervix.

Line 126: cervical cancer Table 1: it is uterus, not cervix.

Line 128: Tables 2, 3, 4, 5. 6, 7 …they explain themselves. Break the single tables and add them one at a time.

Paragraph 3.3.2 It is not clear which comments these comments are referring to.

Line 168: there is no information on the tumor stage.

Line 180: references 11-19: if it were so simple to write an article asking the reader to download the works, read the articles, review and compare them. This work is up to the authors.

The same for references 18,19, 20, 21, 22. And then 23, 24, 25, 26-27-and 28-32.

There are no limits and no strengths.

No comment on what this job really adds.

Author Response

We thank the reviewers for taking time to read our manuscript.

Their suggestions were appropriate, and they were very much appreciated.

We are submitting a revised manuscript, following the suggestions of the reviewers.

REVIEWER 1.

We have hanged the title of the paper, underling that the study analysis includes the pre-pandemic and pandemic COVID19 pandemic.

We have expanded the introduction, trying to clarify the changes associated with the COVID19 pandemic and how these changes may demonstrate the importance of screening for several types of cancer.

We have added more detailed information about the factors involved in screening for breast, colorectal, and cervical cancer.

Several of these limiting factors have been already described, but not in detailed way as in our study.

Probably, quite innovative point in our analysis is the correlation between increased expenditure for preventive care, education and counseling and reduced mortality for colorectal cancer, and cervical cancer.

Probably the increase of expenditure for preventive care is associated with higher implementation of screening programs, but it also associated with other factors like prevention of obesity, control of arterial hypertension; control of lipid levels, improved quality of diet and healthier life style.

For instance, in Switzerland here are no organized screening programs for colorectal cancer screening, and expenditure for preventive care: mortality for colorectal cancer is decreasing more than in other countries. Of course, increased expenditure for preventive care, counseling and education improves education levels of the general population, the possibility of more frequent contacts with family doctors, and more prevention centers for diagnosis and counselling.

W added these points in our manuscript.

The second reviewers asked for a picture defining the trend of the number of operations and screening tests before and during the pandemic. We tried o do, but at the end the picture resulted a little bit confusing and difficult to understand.

We thank the reviewers for their suggestions, which gave us the possibility to clarify several poins in the manuscript, and to improve the quality of our study

Respectfully yours

ANTONIO V STERPETTI, MD

Reviewer 2 Report

Comments and Suggestions for Authors

The following are suggestions to improve the quality of the article:

Add graphs to better appreciate the yearly changes. 

Do not assume that fewer operations and fewer visits with oncologists are harmful; it may be so, but there is also a possibility that we overtreat many patients and mortality may decline with fewer oncology interventions. The data on cancer mortality in most of the years is a major flaw and if possible should be corrected. We would expect a surge in cancer mortality when the epidemic is over but apparently, this is not the case.

.

Author Response

(The authors gave the same response as above.)

Round 2

Reviewer 1 Report

Comments and Suggestions for Authors

The authors have correctly responded to my comments.

As far as I am concerned, the work can be published.

Best regards